# GEM-PA-Based Subunit Vaccines of Crimean Congo Hemorrhagic Fever Induces Systemic Immune Responses in Mice

**DOI:** 10.3390/v14081664

**Published:** 2022-07-28

**Authors:** Qi Wang, Shen Wang, Zhikang Shi, Zhengrong Li, Yongkun Zhao, Na Feng, Jinhao Bi, Cuicui Jiao, Entao Li, Tiecheng Wang, Jianzhong Wang, Hongli Jin, Pei Huang, Feihu Yan, Songtao Yang, Xianzhu Xia

**Affiliations:** 1College of Animal Science and Technology, Shihezi University, Shihezi 832003, China; wq921314@163.com; 2Key Laboratory of Jilin Province for Zoonosis Prevention and Control, Changchun Veterinary Research Institute, Chinese Academy of Agricultural Sciences, Changchun 130122, China; 18203762077@163.com (S.W.); s12k20180101@163.com (Z.S.); zrli20@mails.jlu.edu.cn (Z.L.); zhaoyongkun1976@126.com (Y.Z.); fengna0308@126.com (N.F.); jinhaobi1105@outlook.com (J.B.); liet0706@163.com (E.L.); wgcha@163.com (T.W.); 3Animal Science and Technology College, Jilin Agricultural University, Changchun 130118, China; wjzd2005@163.com; 4College of Veterinary Medicine, Jilin University, Changchun 130062, China; jcc1990512@163.com (C.J.); jin8616771@163.com (H.J.); hp19932015@163.com (P.H.)

**Keywords:** CCHFV, G-GP, subunit vaccine candidates, specific humoral and cellular responses, neutralizing antibody

## Abstract

The Crimean Congo Hemorrhagic Fever Virus (CCHFV) is a tick-borne bunyavirus of the Narovirus genus, which is the causative agent of Crimean Congo Hemorrhagic Fever (CCHF). CCHF is endemic in Africa, the Middle East, Eastern Europe and Asia, with a high case-fatality rate of up to 50% in humans. Currently, there are no approved vaccines or effective therapies available for CCHF. The GEM-PA is a safe, versatile and effective carrier system, which offers a cost-efficient, high-throughput platform for recovery and purification of subunit proteins for vaccines. In the present study, based on a GEM-PA surface display system, a GEM-PA based vaccine expressing three subunit vaccine candidates (G-GP, including G-eG_N_, G-eG_C_ and G-NAb) of CCHFV was developed, displaying the ectodomains of the structural glycoproteins eG_N_, eG_C_ and NAb, respectively. According to the immunological assays including indirect-ELISA, a micro-neutralization test of pseudo-virus and ELISpot, 5 μg GPBLP_3_ combined with Montanide ISA 201VG plus Poly (I:C) adjuvant (A-G-GP-5 μg) elicited GP-specific humoral and cellular immunity in BALB/c mice after three vaccinations via subcutaneous injection (s.c.). The consistent data between IgG subtype and cytokine detection, ELISpot and cytokine detection indicated balanced Th1 and Th2 responses, of which G-eG_N_ vaccines could elicit a stronger T-cell response post-vaccination, respectively. Moreover, all three vaccine candidates elicited high TNF-α, IL-6, and IL-10 cytokine levels in the supernatant of stimulated splenocytes in vitro. However, the neutralizing antibody (nAb) was only detected in A-G-eG_C_ and A-G-eG_C_ vaccination groups with the highest neutralizing titer of 128, suggesting that G-eG_C_ could elicit a stronger humoral immune response. In conclusion, the GEM-PA surface display system could provide an efficient and convenient purification method for CCHFV subunit antigens, and the G-GP subunit vaccine candidates will be promising against CCHFV infections with excellent immunogenicity.

## 1. Introduction

Crimean-Congo Hemorrhagic Fever (CCHF) is a tick-borne zoonotic disease and has spread to more than 30 countries in Africa, Europe, Asia, the Middle East and Europe [1], with obvious locality, seasonality, and sporadic cases. CCHF is caused by Crimean Congo Hemorrhagic Fever Virus (CCHFV), which is a member of the *Nairoviridae* family in the *Bunyavirales*, and its genome comprises three single negative-stranded RNA segments that are known as small (S), medium (M) and large (L) segments, respectively. The small (S) segment encodes the nucleocapsid protein (NP) and the nonstructural protein NSs [2], the M segment encodes a polyprotein precursor (GPC), the large RNA segment (L) encodes an RNA dependent RNA polymerase [1]. The GPC is processed by cellular proteases during maturation, which are cleaved into two structural surface glycoproteins (G_N_ and G_C_) and three other domains: a variable mucin-like domain (MLD), a GP38 domain, and an Nsm domain. The G_N_ gene segment is flanked by the conserved cleavage sites RRLL and RKLL; the G_C_ gene segment starts with RKPL [3], they bind target cells and influence infectivity [4] and cell tropism, and are the targets for neutralizing antibodies [5]. Therefore, G_N_ and G_C_ are the important structural proteins for vaccine development. CCHFV can be transmitted to humans or animals through the bites of infected ticks, directly contacting with contaminated tissues, secretions, blood of infected animals or humans [1], and nosocomial infections [6]. According to the serum positive rate of CCHFV-specific antibodies in wild and domestic animals [7], CCHFV infects a wide range of hosts [8]. There were also cases of transmission from wild and domestic animals to humans [9,10,11,12,13]. Animals infected with CCHFV often remain asymptomatic except for suckling mice, even with high viremia. Infected humans develop severe illnesses, with a case fatality rate of up to 50%. Therefore, CCHFV is a zoonotic potential virus, with wild and domestic animals playing an important role in the risk of transmission to humans. Due to its expanding range, high fatality rate and the lack of an approved vaccine, up to 2017, CCHFV has been designated as one of ten priority emerging infectious diseases by the World Health Organization; and in 2018, the National Institute of Allergy and Infectious Diseases (NIAID) classified the virus as the Category A Priority Pathogen, together with Hantaviruses and Rift Valley Fever within the family *Bunyaviridae*.

The only vaccine candidate which entered human clinical trials was an inactivated vaccine, which was used since 1974 to immunize mainly military and medical personnel, farmers, and persons living or working in endemic regions in Bulgaria [14,15]. Since it was originated from mouse neural tissue content, it was likely to cause autoimmune and allergic responses, so the vaccine was discontinued later. Even though there are currently no licensed vaccines for either humans or animals, researchers have developed a number of approaches for CCHFV, and several vaccine candidates have been demonstrated variable efficacy in multiple animal models for CCHFV, such as whole inactivated virus and cell culture [14,15,16,17], DNA [18,19,20], viral subunits [21], transgenic plant [22], virus-like particle (VLP) [20,23] and mRNA [24]. In addition, viral vectors were used in vaccine development, such as modified vaccinia virus Ankara (MVA) [25,26], human adenovirus type 5 (Ad5) [27], Vesicular Stomatitis Virus (VSV) [28], and bovine herpesvirus type 4 Vector (BoHV-4) [29]. 

Subunit protein vaccines, as the first choice for developing a safe CCHFV vaccine candidate, offer many advantages, such as high product purity, minimal side effects, high safety and yield, and low production costs. At present, there are some subunit protein vaccines about expressions of G_N_ or G_C_ in *Drosophila* Schneider 2 (S2) or *Spodoptera frugiperda* 9 (Sf9) cells that are purified by using His or Strep-Tactin^®^ Superflow^®^ High-Capacity. However, these purification methods are cumbersome in operation and have low yield problems, so they are not promising for large-scale purification. In order to purify the protein conveniently and efficiently, we used an affinity purification method based on a gram-positive enhancer matrix-protein anchor (GEM-PA) surface display system, in which *Lactococcus lactis* (*L. lactis*) was inactivated by hot acid, all intracellular and cell wall components were removed, leaving the rigid peptide-glycan (PGN) matrix remaining [30]. Treated *L. lactis* were also called GEM particles (GEMs) that were non-living and have a diameter of approximately 1–2 um [31]. An antigen may be expressed with a peptidoglycan anchoring domain (PA) as a fusion protein in different expression systems; the fusion protein can be purified by noncovalently attached to the peptidoglycan surface of GEMs through PA, generating a GEM-PA-antigen complex [32]. *L. lactis* is considered as safe by the Food and Drug Administration (FDA), and the efficacy of the GEM-PA display system has been demonstrated in various protein vaccine candidates including antigens of parasitic, viral, or bacterial antigens [32,33,34,35,36,37,38,39,40,41,42,43,44]. Therefore, the GEM-PA surface display system could provide an efficient and simple purification method for CCHFV subunit antigens. 

In this study, we used the GEM-PA display system to display the eG_N_, eG_C_ and conserved neutralizing body epitope (NAb, aa1443 and 1566 of the M gene in IbAr10200 strain) of glycoprotein and studied the immunogenicity in BALB/c mice. This study provides an efficient and convenient purification method for CCHFV subunit antigens; the subunit vaccine was termed G-GP (including G-eG_N_, G-eG_C_, G-NAb), which represents a promising vaccine candidate against CCHFV infections.

## 2. Materials and Methods

### 2.1. Construction and Expression of Recombinant Baculoviruses

In this study, three truncated fragments of CCHFV Ibar10200 GP (GenBank ID: AF467768) were designed, namely the extracellular region of mature G_N_ (eG_N_,520-696aa), the extracellular region of G_C_ with truncation of C-terminal 19aa (eG_C_,1041-1579aa) and the conservative neutralizing antibody region of G_C_ (NAb,1443-1566aa). To improve the expression of recombinant CCHFV GP, all genes were codon optimized according to the specific codon usage of Sf9 cells. Target protein genes were fused at the amino terminus to the gp67 signal sequence and were co-expressed with a gene encoding PA_3_ through a 24-amino acid glycine linker (GGTGGTTCTGGTGGTGGTTCTGGT), respectively. PA_3_ genes encoding three LysMs were amplified by PCR using codon-optimized pUC57-PA_3_ (*L. lactis* MG1363) as a template with the oligonucleotide primers linker-PA_3_-F and PA_3_-R. Followed by a 6 × His-tag (CATCACCATCACCATCAC) for the easy detection of protein expression, KpnI cleavage site was added at the 5′ end of the construct, cleavage site for XbaI was designed at 3′ termini. The GPLP_3_ (including eG_N-_LP_3_, eG_C_-LP_3_, NAb-LP_3_) fusion gene was amplified by PCR using synthetic oligonucleotide primers as listed in Table 1 and the amplified segments were inserted into the baculovirus expression vector pFastBac1-gp67 (Invitrogen, Carlsbad, CA, USA), respectively. The successfully constructed plasmids (pFastBac1-gp67-eG_N_-LP_3_, pFastBac1-gp67-eG_C_-LP_3_, pFastBac1gp67-NAb-LP_3_) were then transformed into Escherichia coli DH10Bac competent cells to generate recombinant bacmids (rBacmid-eG_N_LP_3_, rBacmid-eG_C_LP_3_, rBacmid-NAbLP_3_). Sf9 insect cells (Thermo Fisher Scientific, Waltham, MA, USA) were cultured in TNM-FH insect medium (Sigma, St. Louis, MO, USA) and supplemented with 10% fetal bovine serum (FBS, Gibco, Grand Island, NY, USA) and 1% Penicillin-Streptomycin (Caisson, Grand Island, NY, USA). According to Cellfectin^TM^ II Reagent (Invitrogen, Waltham, MA, USA) and Bac-to-Bac Expression Systems manual, we transfected several 2 ug of validated bacmids (rBacmid-eG_N_LP_3_, rBacmid-eG_C_LP_3_, rBacmid-NAbLP_3_) into 2 × 10^6^ cells/mL Sf9 cells that were seeded beforehand in 6-well plates. 4 days post transfection (p.t.), cell debris were removed and three kinds of supernatant containing proteins were collected, which were defined as the first passage1 (P1) of recombinant baculoviruses (rBVs), including rBV-eG_N_LP_3_, rBV-eG_C_LP_3_, rBV-NAbLP_3_. P1rBVs were expanded in Sf9 cells that were incubated in a cell culture dish with a 100 nm × 20 mm style (Corning-Costar, Corning, NY, USA) at 27 °C, after 96 h post infection (p.i.), passage 2 (P2) rBVs were harvested, and so on, until the generation of P3 rBVs.

### 2.2. Expression of CCHFV GPLP_3_ Fusion Protein

The expression of exogenous genes was identified by indirect immunofluorescence assay (IFA) and the expression forms of proteins were assessed by western blot (WB).

For IFA, Sf9 cells were infected with P2 rBVs and were cultured in 24-well plates for 48 h, then fixed in 80% cold acetone for 30 min at room temperature and washed three times with phosphate buffer saline (PBS containing 0.05% Tween 20, PBST), then cultured plates were incubated with a mouse polyclonal anti CCHFV-G_N_, a mouse mAb clone 11E7anti-CCHFV pre-Gc, a mouse mAb clone 11E7 anti-CCHFV pre-Gc (prepared and stored in our laboratory; BEI Resources, Manassas, VA, USA; BEI Resources, Manassas, VA, USA, with a 1:500 or 1:1000 dilution containing 1% bovine serum albumin, BSA) at 37 °C for 1 h, after three washing steps, an FITC-labeled goat against mouse IgG antibody with a 1:200 dilution (Bioworld, Minnesota, MN, USA) was added with Evans blue with a 1:100 dilution (Sigma, St. Louis, MO, USA) for 1 h at 37 °C. After repeating the washing steps, results were observed by the fluorescence microscope (Zeiss, Oberkochen, Germany).

For WB analysis, cell cultures infected with P2 rBVs were centrifuged at 6500× *g* at 4 °C for 10 min, cell supernatants were directly collected, precipitates were then resuspended in the same volume of 0.01 M PBS (pH 7.2–7.4) and crushed by ultrasound, after SDS-PAGE under denaturing conditions, prepared samples were transferred onto a nitrocellulose (NC) membrane (GE Healthcare Life Sciences, Freiburg, Germany) for immunoblot with a mouse anti-CCHFV-GP antibody (anti-CCHFV-G_N_ polyclonal antibody, a mouse mAb clone 11E7anti-CCHFV pre-Gc, a mouse mAb clone 11E7anti-CCHFV pre-Gc), and an HRP-conjugated secondary antibody (Bioworld, Minnesota, MN, USA).

### 2.3. Production of GEM and binding GPLP_3_ proteins to Immunobiotic G-GP

GEM particles were prepared as described previously [45]. Briefly, overnight cultures (100 mL) of *L. lactis* MG1363 in M17 Broth (Hopebiol, Qingdao, China) were harvested by centrifugation (10 min, 12,000× *g*) and washed with PBS, precipitates were then boiled in 10% trichloroacetic acid (Sigma, St. Louis, MO, USA) for 30 min, followed by extensive washing with PBS five times. Finally, treated GEMs were re-suspended in sterile PBS (pH 7.2–7.4), with a Bürker-Turk counting chamber (Lab Unlimited, UK); the number of GEMs per milliliter was counted. Finally, one unit (U) was defined as 2.5 × 10^9^ GEM particles, and the GEMs were stored at −80 °C until use.

Cell-free antigen (eG_N_LP_3_, eG_C_LP_3_, NAbLP_3_) of culture supernatant was separately bound to the surface of the formulated GEMs (one U of GEM particles were added into 10 mL of each rBVs culture supernatant, in contrast, poor solubility of the eG_C_LP_3_ protein, which must be broken by ultrasound to obtain sufficient protein. Following by gentle agitation for 1 h at room temperature and a centrifugation step (8 min, 6500× *g* at 4 °C), precipitates (the bound fractions) were washed five times to eliminate unbound proteins, collected proteins were stored in sterile PBS (pH 7.2–7.4, 0.01 mM) at a concentration of 200 μL/U at −20 °C until further use. 

The capacity of this purification method was explored. 0.5 U of GEMs separately mixed 0–10 mL of treated GPLP_3_ (eG_N_LP_3_, eG_C_LP_3_, NAbLP_3_) supernatant proteins, formulated G-GP (G-eG_N_, G-eG_C_, G-NAb) in 10 mL crude protein-containing extract were calculated with using the Pierce BCA protein assay kit (Thermo Fisher Scientific, Waltham, MA, USA) and following the formula: Content _GPLP__3_ = Content _G_-_GP_- Content _GEM_.

### 2.4. SDS-PAGE, Western Blot and Immuno-Electron Microscope (IEM) Identification of the Binding GEM Particles

Surface display of GPLP_3_ proteins on GEM particles (G-GP) was verified by SDS-PAGE, Western Blot, and the immuno-electron microscope (IEM).

For the SDS-PAGE and WB identification of the binding GEM particles, the GPBLP_3_ was resuspended in 5 × SDS-PAGE sample buffer (Beyotime Biotechnology, Shanghai, China), separated by 10% SDS-PAGE and then transferred by electroblotting onto NC transfer membranes under denaturing conditions for Western blotting with a mouse anti-CCHFV-G_N_ specific for CCHFV-G_N_ polyclonal antibody (diluted 1:500), a mouse mAb clone 11E7 anti-CCHFV pre-Gc (diluted 1:1000; BEI Resources, Manassas, VA, USA), a mouse mAb clone 11E7 anti-CCHFV pre-Gc (diluted 1:1000, BEI Resources, USA), and an HRP-conjugated secondary antibody (diluted 1:5 000; Bioworld, Minnesota, MN, USA).

For IEM, a mouse antiserum specific for CCHF-G_N_ (diluted 1:200), mouse mAb clone 11E7 anti-CCHFV pre-Gc (diluted 1:1000), a mouse mAb clone 11E7 anti-CCHFV pre-Gc (diluted 1:1000), the immuno-electron microscope anti-Mouse IgG (whole molecule) Gold antibody produced in goat (diluted 1:200, Sigma, St. Louis, MO, USA) were used. Samples were observed under the electronic microscope.

### 2.5. Immunization of Mice

To evaluate the immunogenicity of the G-GP (G-eG_N_, G-eG_C_, G-NAb) vaccine candidates against CCHFV Negeria-IbAr10200 strain in mice, a total of 75 female BALB/c (6–8 week-old) mice were randomly divided into 15 groups (Table 2), nine groups of BALB/c mice (*n* = 5 per group) subcutaneously received different doses of A-G-GP (A-G-eG_N_, A-G-eG_C_, A-G-NAb), namely G-GP (1, 5, or 20 μg dose) was adjuvanted with a complex of ISA201VG (Seppic, Paris, France) and PolyI:C (Sigma, St. Louis, MO, USA), respectively. Another three groups of BABL/c mice (*n* = 5 per group) subcutaneously received a 5 μg dose of the G-GP vaccine candidate alone (G-eG_N_, G-eG_C_, G-NAb). The final three control groups of BABL/c mice (*n* = 5 per group) were subcutaneously immunized with GEM + 201 + polyI:C, 201 + polyI:C or PBS. Booster doses were administered 3, 6 and 9 weeks after the initial immunization. Sera were collected via submandibular bleeds 2 weeks after each post-immunization (2, 5, 8, 11 wpi) and inactivated at 56 °C for 30 min before stored at −80 °C until further use in the indirect-ELISA and virus neutralization assays.

### 2.6. Virus-Specific IgG and Subtype Detection

To detect the potential of our G-GP constructs to induce specific IgG antibody responses in BALB/c mice, we developed an indirect enzyme linked immunosorbent assay (ELISA) to quantify the specific IgG and subtype antibody levels (IgG1, IgG2a) in serum from mice vaccinated with 5 μg A-G-GP and 5 μg G-GP, following the steps described below. Meanwhile, we also used the ratio of IgG2a to IgG1 as an indirect method to evaluate induced T-helper 1 (Th1) and T-helper 2 (Th2) biases, respectively. 

Briefly, recombinant eG_N_ and NAb of the CCHFV specific glycoprotein expressed in a prokaryotic system were used as ELISA antigens, which were diluted in 0.05 M carbonate-bicarbonate buffer pH 9.6 (Sigma, St. Louis, MO, USA), and used to coat Maxisorp 96-well plates (Corning-Costar, Corning, NY, USA) at 100 and 400 ng/well in a volume of 100 μL/ well, respectively. Plates were covered with plate sealers and incubated overnight at 4 °C. The next day plates were washed with PBST (PBS containing 0.05% Tween-20) for three times and blocked with 150 μL/well of 3% bovine serum albumin in ddH_2_O (3%BSA, Sigma, St. Louis, MO, USA) for 1 h at 37 °C. After plates were washed again, serum samples from immunized BALB/c mice were serially diluted two-fold starting at a dilution of 1:80 in 1% BSA buffer, and added to the wells. Finally, the plates were incubated at 37 °C for 1.5 h. After 3 rounds of wash, anti-mouse IgG-HRP antibody (Bioworld, St. Louis, MO, USA) at a dilution of 1:10,000, anti-mouse IgG1-HRP and anti-mouse IgG2a-HRP (Southern Biotech, Birmingham, AL, USA) at 1:5000 dilution was separately added to each well and further incubated for 1.5 h at 37 °C. Finally, plates were washed three times with PBST, and incubated for 5~10 min with tetramethyl l benzidine substrate (TMB, Sigma, St. Louis, MO, USA) at room temperature. Subsequently, the reaction was stopped by adding 0.5 M H_2_SO_4_ (Sigma, St. Louis, MO, USA), the plates read at 450 nm in an ELISA reader (Bio-Rad, Hercules, CA, USA). For data analysis, titers were determined as the highest dilution at which the mean absorbance of the sample was 2-fold greater than the mean absorbance of the same dilution of control serum, additional analysis was carried out using GraphPad Prism 8.0. software (GraphPad Software Inc., San Diego, CA, USA). 

### 2.7. Neutralization Assay

Handling of CCHFV requires high-containment facilities, biosafety level 3 (BSL-3) and BSL-4 facilities in endemic and non-endemic areas, respectively. To overcome this limitation, we generated a pseudo-typed virus-CCHF_VPV_, in which the G gene of VSV was replaced by the CCHFV-GP (IbAr10200 strain), an enhanced green fluorescent protein (eGFP) reporter gene was cloned between the N gene and P gene of the full-length genome cDNA of VSV [46]. CCHF_VPV_ titration was performed on 10-fold serial dilutions (prepared in DMEM) of transfected cell supernatants, using a TCID_50_ assay on Vero E6 cells in 96-well plates. Plates were incubated with the supernatants 48 h at 37 °C, and were observed under a fluorescence microscope. CCHF_VPV_ titration was calculated using the Reed and Muench formula, and expressed as TCID_50_ per mL of stock.

CCHF_VPV_ system was used to evaluate the potential of neutralizing activity in sera from mice immunized with G-GP (G-eG_N_, G-eG_C_, G-NAb). Mice sera were diluted in a 2-fold dilution series, with 4-fold in the initial dilution, and mixed with an equal volume of CCHF_VPV_-containing supernatant and incubated for 1–2 h at 37 °C. The mixture was then applied to Vero E6 cells cultured in a 96-well plate. Meanwhile, Anti-CCHFV pre-GC mAb clone 11E7 (BEI Resources, Manassas, VA, USA), anti-control serum of mouse and CCHF_VPV_ in different dilutions were set as the positive, negative antibody and antigen control, respectively. 11E7 was diluted in a 2-fold dilution series (1:32 to 1:16384-fold dilution). Plates were incubated for 48 h at 37 °C and observed with a fluorescence microscope. Data were analyzed as previously reported using GraphPad Prism 8.0. software (GraphPad Software Inc., San Diego, CA, USA). 

### 2.8. IFN-γ and IL-4 Cytokine Detection

Specific T cell responses were detected by the quantification of IFN-γ and IL-4 production from splenocytes by ELISpot. Groups of 3 BALB/c mice were vaccinated subcutaneously with 5 μg either the A-G-GP (A-G-eG_N_, A-G-eG_C_, A-G-NAb) vaccines or G-GP (G-eG_N_, G-eG_C_, G-NAb) vaccines for three times. 2 weeks after the third immunization (8 wpi), the plate was blocked with RPMI 1640 medium (Gibco, Grand Island, NY, USA) for 1 h before the addition of the splenocytes. 2.5 × 10^5^ freshly isolated mouse splenocytes were added to each well, and then stimulated with purified eG_N_ or NAb antigen produced in *E. coli*, PBS was used as a negative control. Splenocytes incubated in culture media alone served as a background control and splenocytes stimulated with 10 μg/mL of concanavalin A (Sigma, St. Louis, MO, USA) as a cell viability control.

The splenocytes were stimulated and cultured at 37 °C for 24 h. IFN-γ and IL-4 were detected by using mouse enzyme-linked immune-spot (ELISpot) kits (MABTECH, Nacka, Sweden) according to the manufacturer’s instructions. Spot-forming cells (SFCs) were counted with an ELISpot reader (AID ELISPOT reader-iSpot, AID GmbH, GER).

### 2.9. Cytokine Measurement of Splenocyte Culture Supernatants

Mouse ELISA cytokine kits (MABTECH, Nacka, Sweden) were applied to detect the levels of the cytokines Th1 (IFN-γ, TNF-α, IL-2) and Th2 (IL-4,6,10) in supernatants of stimulated splenocytes, which were harvested 14 days after the third immunization from mice immunized with 5 μg A-G-GP or 5 μg G-GP; collected splenocytes were spread at 2.5 × 10^5^ cells/well into 96-well plates and were stimulated with purified eG_N_ or NAb antigen (10 μg/mL) produced in *E. coli* for 72 h at 37 °C and 5% CO_2_. According to the manufacturer’s instructions, IL-2,4,6,10, IFN-γ, and TNF-α levels in supernatants were analyzed.

### 2.10. Data Analysis

The results are expressed as the means ± SD. Figures were generated using GraphPad Prism 8.0. software (GraphPad Software Inc., San Diego, CA, USA). Significance differences between the groups were analyzed using one-way ANOVA or two-way ANOVA were deemed significant at *p* values of 0.05 or less. Statistical significance is indicated as * *p* < 0.05, ** *p* < 0.01, *** *p* < 0.001, and **** *p* < 0.0001.

## 3. Results

### 3.1. Generation of Recombinant Baculovirus and Expression of CCHFV-eG_N_LP_3_, CCHFV-eG_C_LP_3_ and CCHFV-NAbLP_3_ Fusion Proteins 

A schematic representation of the glycoprotein ORF of CCHFV Ibar10200 strain was shown in Figure 1A. We designed three GPLP_3_ (eG_N_LP_3_, eG_C_LP_3_, NAbLP_3_) fusion proteins, as shown in Figure 1B, in which eG_N_, eG_C_-19aa and NAb gene were fused to PA_3_ gene with a linker, respectively. IFA results showed that rBV-GPLP_3_ (rBV-eG_N_LP_3_, rBV-eG_C_LP_3_, rBV-NAbLP_3_) infected Sf9 cells fluoresced green (Figure 1(D5–D7)), compared with the control cells (Figure 1(D8)), suggesting that three exogenous genes were successfully expressed. Western blot analysis also proved that three fusion proteins were successfully expressed, in which two recombinant proteins (eG_N_LP_3_, NAbLP_3_) were secreted successfully into the supernatants, respectively, presenting as a soluble 55 KDa and 50 KDa protein boxed out in red (Figure 1(D9,D11)), while eG_C_LP_3_ had poor solubility, which must be broken by ultrasound to obtain sufficient protein, presenting as a 100 KDa protein boxed out in red (Figure 1(D10)).

### 3.2. Location of Fusion Proteins on GEM Particles

Schematic representation of GP-PA binding with GEM particles was shown in Figure 2A. The surface location of GPLP_3_ (eG_N_LP_3_, eG_C_LP_3_, NAbLP_3_) fusion proteins on GEM particles, which were confirmed by SDS-PAGE, Western blot and immunogold electron microscopy (IEM). 

Western blot analysis with antibody demonstrated that binding to the 55KDa, 100 KDa and 50 KDa were recombinant proteins, and no expression of any immune reactivity protein was detected in mock preparations (Figure 2(B1,B2)). Analysis by SDS-PAGE revealed that three recombinant proteins with the expected apparent molecular weight of approximately 55 KDa, 100 KDa and 50 KDa, respectively (Figure 2(C1–C3)). All target proteins are boxed out in red. In addition, IEM analysis exhibited that the elliptical shape of GEM particles could be clearly observed with evident surface gold particles, which strongly indicated that the G-GP (G-eG_N_, G-eG_C_, G-NAb) fusion proteins successfully were displayed on the surface of GEM particles (Figure 2(D6–D8)).

In this experiment, SDS-PAGE and Western blot analysis showed that three G-GP proteins were observed in precipitates recovered after incubation with GEM particles. The IEM results were consistent with Western blot analysis, suggesting binding of fusion proteins on GEM particles did not affect the immune reactivity.

### 3.3. Binding Activity of Fusion Proteins on GEM Particles

To analyze the maximum binding capacity of each fusion protein on the GEM particles, 0.5 U GEM particles were mixed with 0, 2, 4, 6, 8 and 10 mL of each recombinant baculovirus culture supernatant or supernatant after ultrasonic crushing; after washing, three mixtures were prepared into samples, respectively. The results were displayed in Figure 3A–C, with the increase in binding culture supernatant volume, the relative quantity of binding fusion proteins (G-eG_N_ and G-NAb) increased. The binding capacity of G-eG_C_ fusion protein increased, when 0.5 U GEM particles were combined with 8 and 10 mL of recombinant baculovirus culture supernatant after ultrasonic crushing. All relative maximum binding capacity of 0.5 U GEM particles were 10 mL volumes, which was consistent with descriptions in other reports [33,34]. Furthermore, we estimated that 1 U GEM particles can bind 280 μg eG_N_LP_3_ fusion protein, 220 μg eG_C_LP_3_ fusion protein and 260 ug NAbLP_3_ fusion protein with using the Pierce BCA protein assay kit. Thus, a yield of approximately 14 mg eG_N_LP_3_, 11 mg eG_C_LP_3_, 13 mg NAbLP_3_ proteins were obtained from three baculovirus cultures per liter, respectively.

### 3.4. Virus-Specific IgG and Subtype

To determine the antibody responses of G-GP (A-G-eG_N_, A-G-eG_C_, A-G-NAb) vaccines against CCHFV-GP, BALB/c mice were immunized subcutaneously with the three vaccine preparations containing 1 μg, 5 μg, 20 μg for four times; the sera in each subgroup were collected at 2 weeks after each immunization and analyzed by indirect-ELISA. 

As expected, the IgG antibody in the control groups were below the limit of detection from 2 wpi to 11 wpi (Figure 4A–C). The first immunization of mice with G-GP vaccines containing 1 μg, 5 μg and 20 μg dose groups did not generate any detectable IgG antibodies at 2 wpi (Figure 4A–C and Table 3). Mice in all G-GP (1 μg, 5 μg and 20 μg) vaccination groups did not generate detectable IgG antibodies at 2 wpi post first-dose vaccination (Figure 4A–C and Table 3). The booster vaccination with G-eG_N_, G-eG_C_ and G-NAb dramatically increased the IgG antibodies, with the geometric mean titer (GMT) reaching (G-eG_N_1536,3072,3584; G-eG_C_ 2048,5120,1126.4; G-NAb 2304, 3584, 4096) in 1 μg, 5 μg and 20 μg dose vaccination groups, respectively (Figure 4A–C and Table 3). 5 μg and 20 μg dose groups showed relatively high levels of anti-CCHFV GP IgG titer and significantly higher than that in the 1μg dose A-G-GP group after three immunizations, whereas there was no significant difference between the 20 μg dose A-G-GP group and 5 μg dose A-G-GP group (Figure 4A–C). Furthermore, a fourth immunization was added. However, IgG antibody levels in all sera from A-G-GP immunized mice showed no significant differences between 8 wpi and 11 wpi (Figure 4A–C), suggesting IgG antibody titers were close to saturation after the third immunization. Therefore, three vaccinations with a 5 μg dose A-G-GP were selected for further vaccination experiments in mice, which were sufficient to produce IgG antibodies, reaching endpoint GMT of IgG up to 22,528, 20,480, 22,528, respectively (Figure 4A–C and Table 3).

The subtypes of CCHFV-specific IgG antibodies were detected in 5 μg A-G-eG_N_, A-G-eG_C_, A-G-NAb and 5 μg G-eG_N_, G-eG_C_, G-NAb groups. The antibody subtype could reflect the bias of Th1 and Th2 responses. All immunized mice clearly responded with IgG1and IgG2a, a similar phenomenon to IgG antibodies was found for CCHFV specific IgG1 antibodies, which were detected in high levels at 5 wpi and increased significantly at 8 wpi, and there were no significant differences between 8 wpi and 11 wpi, advising IgG1 antibody titer was saturated in the collected sera from A-G-GP-5 μg immunized mice (Figure 4D–F). Interestingly, IgG1 antibody titer in 5 μg G-eG_N_ and G-NAb alone group at 11 wpi were significantly higher than those at 8 wpi, whereas the phenomenon did not appear in the 5 μg G-eG_C_ group (Figure 4D–F). Specific IgG2a antibodies were detectable in the sera at a lower level than IgG1 at 5 wpi, and were found to have increased dramatically at 8 wpi and reached the highest level at 11 wpi (Figure 4D–F), indicating IgG2a had a delay with respect to IgG1. We also detected all the ratios of IgG2a/IgG1of sera in 5 μg A-G-GP and 5 μg G-GP alone group and found that all the ratios of IgG2a/IgG1of sera in 5 μg A-G-GP and 5 μg G-GP alone group increased and significantly increased from the 5 wpi to 11 wpi, with the times of immunizations increasing (Figure 4G–I), indicating a process of reaching a balanced Th1 and Th2 immune response. 

Altogether, both A-G-GP and G-GP could induce a mixed Th1 and Th2 immune response after three vaccinations, which are considered to be important correlates of protection [20]. 201 + polyI:C adjuvant could significantly enhance G-GP to produce IgG1 antibody responses (Th2), but repeated doses were required to induce IgG2a antibodies against CCHFV. In addition, the increased IgG2a and IgG2a/IgG1 ratio reflected a strengthened Th1 response in all the immunized BALB/c mice except the control group (Figure 4G–I).

### 3.5. Virus Neutralization Assay

To evaluate the nAbs developed in all vaccination groups, a CCHF_VPV_ neutralization assay in a BSL-2 setting was established and applied. The first immunization of mice with the eG_C_BLP_3_ vaccine containing 1 μg, 5 μg and 20 μg dose groups did not develop any detectable neutralization antibodies (Figure 5 and Table 3). After the second immunization, the 1, 5, and 20 μg dose groups showed neutralizing antibody titers of 1:5.6, 1:16 and 1:19.2, respectively, against CCHFV_VPV_ (Figure 5 and Table 3). The third immunization led to significantly elevated levels of neutralizing antibody titers in the 5 μg and 20 μg dose groups, with the neutralizing antibody titers of 1:70.4 and 1:76.8, respectively (Figure 5 and Table 3), while neutralizing antibody titers in the 1 μg dose group remained at a low level, 1:14.4 (Figure 5 and Table 3). These results advised that the neutralizing antibody titer elicited by the G-eG_C_ vaccine increased in a dose-dependent manner. There was no significant difference in the CCHFV-specific neutralizing response between three and four immunizations with A-G-eG_C_-5 μg, 20 μg and A-eG_C_-5 μg, 20 μg (Figure 5 and Table 3), suggesting the neutralizing antibody levels from mice immunized with A-G-eG_C_-5 μg gradually increased to plateau at 8 wpi (Figure 5 and Table 3), and the addition of 201 + polyI:C adjuvant did not significantly increase the neutralizing antibody titers in three immunological doses group. Interestingly, compared with the neutralizing capacity of sera in 1 μg soluble G-eG_C_ group, sera in all the G-eG_N_ and G-NAb groups were not sufficient to neutralize pseudo-virus infection in Vero E6 cells, albeit very low (Figure 5 and Table 3), which was inconsistent with the study that aa 1443 and 1566 of the M gene in Ibar10200 strain was the conserved neutralizing epitope. At the same time, our results demonstrated that the mAb 11E7 were able to neutralize the pseudo-virus infection in Vero E6 cells (the neutralizing titer up to 1:512), and sera in control groups were unable to block pseudo-virus infection in Vero E6 cells, suggesting that the CCHF_VPV_ system was effective in neutralization test.

In a word, the results showed that all of the CCHFV-G-eG_C_ vaccinated BALB/c mice developed detectable IgG and neutralization antibodies to CCHFV-GP after two-dose immunizations (Figure 5 and Table 3), indicating G-eG_C_ could elicit stronger humoral immune response, but non-neutralization antibody response was detected in all serum samples from mice vaccinated with G-eG_N_ or G-NAb.

### 3.6. Antigen-Specific Cellular Immune Responses 

To further investigate antigen-specific cellular immune responses, the secretion of IFN-γ and IL-4 by mouse splenocytes was measured by ELISpot assay. In Figure 6A,B, the ELISpot assay showed that the amount of splenocytes secreting IFN-γ and IL-4 from mice immunized with 5 μg A-G-GP or 5 μg G-GP alone was significantly higher than from mice in the control group, suggesting 5 μg G-GP could increase in production of this cytokine. Additionally, the secretion of IFN-γ and IL-4 by splenocytes from mice in A-G-GP group were significantly strong than that from mice in G-GP group, indicating that the 201 + polyI:C adjuvant could significantly enhance cellular immune responses. In addition, the A-G-eG_N_ and G-eG_N_ vaccines were capable of producing significantly higher numbers of IFN-γ and IL-4 SFCs, when compared with the numbers observed for the A-G-eG_C_, G-eG_C_, A-G-NAb and G-NAb vaccines (Figure 6A,B), suggesting the A-G-eG_N_ and G-eG_N_ vaccines could elicit a stronger T-cell response. Together, our ELISpot data demonstrated that the G-GP vaccine induced CCHFV-specific IFN-γ and IL-4 recall responses against three vaccine antigens. As a negative control, we included naive splenocyte cells from GEM + 201 + polyI:C immunized negative control mice.

### 3.7. Cytokine Secretion by Restimulated Splenocytes

Th1 (IFN-γ, TNF-α, IL-2) and Th2 (IL-4, IL-6, and IL-10) cytokines in the supernatants from splenocytes ex vivo restimulated with eG_N_ or NAb were detected by ELISA. 

As the Figure 7A–F showed, the levels of the Th1 (IFN-γ, TNF-α, IL-2) and Th2(IL-4, IL-6, IL-10) cytokines secreted by splenocytes from immunized with 5 μg A-G-GP (A-G-eG_N_, A-G-eG_C_ or A-G-NAb) or the 5 μg G-GP alone group were significantly higher than those from the mice in the control group, suggesting 5 μg dose G-GP resulted in significant increases in the secretion of these cytokines, and that these levels were also significantly enhanced in the presence of the ISA 201VG plus Poly (I:C) adjuvant. The A-G-eG_N_ and G-eG_N_ vaccines were capable of producing significantly higher levels of cytokine, when compared with the numbers observed for the A-G-eG_C_, G-eG_C_, A-G-NAb and G-NAb vaccines (Figure 7A–F), suggesting that the A-G-eG_N_ and G-eG_N_ vaccines could elicit a stronger Th1 and Th2 immune response. In addition, the results showed that the potential to stimulate the production of adequate amounts of TNF-α, IL-6, IL-10 in the supernatant samples (Figure 7B,E,F), and the G-GP (G-eG_N_, G-eG_C_ or G-NAb) vaccine enhanced the secretion of both Th1 and Th2 cytokines in splenocytes, when vaccinated with adjuvant (Figure 7A–F). In conclusion, cytokine secretion data demonstrated that the GP-GP vaccine induced a balanced Th1 and Th2 immune response against CCHFV-GP. 

## 4. Discussion

The CCHFV is one of the most geographically widespread tick-borne viruses. Due to high case fatality rates, expanding of endemic regions, and the lack of licensed vaccines or effective therapeutics, potentially increasing risk of transmission from human-to-human, animal-to-animal, and even animal-to-human, CCHFV is a serious threat to public health, which has been added by the World Health Organization to the list of priority pathogens (http://www.who.int/blueprint/priority-diseases/en/, accessed on 18 June 2022). Therefore, establishing reasonable vector control, especially developing safe and effective vaccines against CCHFV for all populations at risk of exposure to CCHFV, is critically important. 

Currently, in addition to employing inactivated viruses, vaccine design of CCHFV is mainly based on delivering two main immunodominant viral genes, namely S and M. Since the exposed position on the virion surface makes it able for antibody binding and neutralization, glycoprotein is often used as the candidate antigenic target in novel vaccine designs. The two structural surface glycoproteins (G_N_ and G_C_) are the focus in our subunit vaccines, as they have good immunogenicity and can elicit virus-neutralizing antibodies. In addition, we also developed a highly conserved and neutralizing antibody (NAb) region of GP (between aa 1443 and 1566 of the M gene in IbAr10200 strain) [47], which could be recognized by mAb11E7 that was identified as an antibody possessing broadly cross-neutralizing abilities [48], and has shown protection against the lethal strain IbAr10200 of CCHFV infection in the suckling mouse [5]. To our knowledge, this is the first related vaccine expressing the NAb region of the glycoprotein. Furthermore, an effective affinity purification method called GEM-PA surface display technology was used in the purification of the expressed proteins. Purity of protein is so important that it is necessary to adopt an efficient, economical and safe purification method. At present, traditional purification methods for protein subunit vaccines mainly rely on His or Strep-tag purification resin [21,49,50], in which many unavoidable problems exist, such as cumbersome operation, low purification efficiency, the need of many expensive purification columns and so on. However, in the GEM-PA surface display system, we just need to combine cultural supernatant or cell lysate with treated *L. lactis* for an hour at room temperature, followed by repeated centrifugation steps and washing with sterile PBS, for us to obtain high purity protein. It is such a convenient and timesaving method that the whole purification process only took 1.5 h, as well as having a high purification efficiency, which could maximize condensed proteins from cultural supernatant or lysate, and a yield of approximately 14 mg protein was obtained from cultures of baculovirus in our study. Furthermore, the method could be easily manipulated for later dilution, storage, and preparation for vaccines, and is economical to scale up. In addition, it is the first time that the GEM-PA surface display system is applied to the purification of CCHFV GP subunit vaccines.

In the present study, we developed three subunit vaccine candidates, namely G-eG_N_, G-eG_C_, G-NAb, which were combined with Montanide ISA 201VG plus Poly (I:C) adjuvant and elicited humoral and cellular immunity against CCHFV. 

ELISA and the pseudo-typed virus neutralization assay were successfully used for the detection of specific IgG and neutralizing activities against the CCHFV glycoproteins of collected mice serum in vitro under BSL-2 containment. The data demonstrated that the variation trend of neutralizing antibody had some consistence with IgG antibody, including the time point of seroconversion, the saturation effect and the dose-dependent manner of immunogenicity (Figure 4A–C and Table 3). The difference was that specific IgG antibodies produced in all the G-GP vaccinated mice, whereas specific neutralizing antibodies were only detected in the group that mice were vaccinated with A-G-eG_C_ and G-eG_C_ vaccine (Figure 5), which were consistent with what has been mentioned in the literature [5], in which only a number of mAbs against G_C_, were able to neutralize the virus infection of SW-13 cells in vitro; none of the mAbs directed against G_N_ exhibited neutralizing activity in plaque reduction assays. In another bunyavirus, La Crosse virus (LACV), whose G_C_ glycoprotein is sufficient to block virus infection in mammalian cells, while antibodies against G_N_ neutralize infection in a mosquito cell line but not in a vertebrate cell line [51]. Therefore, G-eG_C_ could elicit a stronger humoral immune response, and CCHFV neutralization is likely to be related to cell lines that needs more in-depth studies. Regrettably, all the sera from mice immunized with A-G-NAb (1, 5, 20 μg) and G-NAb-5 μg did not demonstrate any neutralization against CCHFV-GP, albeit at low levels. This result was not completely unexpected since NAb protein is made up of just 124 amino acids (1443–1566aa, 124aa), which may be too short for the extracellular region of G_C_ made up of just 558 amino acids (1041–1598aa, 558aa), the single expression of the NAb region may not produce natural conformation structure. Furthermore, we found that the 5 μg A-G-eG_C_ vaccine could produce high levels of IgG antibodies (1:40,960) but low neutralizing antibodies (the highest level was 1:128 in individual mice), even though we immunized the mice four times, which was similar to previously reports, such as neutralization antibodies from the only study on CCHFV in vaccinated individuals (PRNT_50_< 1:32) [14], cell culture based and the mouse brain derived inactivated vaccines in the BALB/c mice (FRNT:114.3, in 20 μg group) [16], the first DNA vaccine in BALB/c mice (PRNT_50_ < 1:40–1:160) [52], and it has been shown to have low neutralizing activity of convalescent-phase serums from patients (titers ranging from 8 to 32) [53]. The reason why the levels of IgG antibodies and neutralizing antibodies differ so much deserves further study.

We also used ELISpot and ELISA assays to detect T-cell responses and the Th1 and Th2 cytokines of the systemic pattern in the splenocyte supernatants, respectively. ELISpot analysis demonstrated a significantly increased IFN-γ and IL-4 T-cell response in all mice vaccinated with A-G-GPand G-GP compared to mice vaccinated with control antigen (Figure 6A,B). The levels of Th1-type biomarkers (IFN-γ, TNF-α and IL-2) and Th2-type biomarkers (IL-4, IL-6, IL-10), were shown to be significantly elevated in all the G-GP vaccine-immunized mice after three immunizations, indicating G-GP induced a balanced Th1 and Th2 immune profile again (Figure 7A–F). Additionally, the secretion of cytokines indicated that the production of adequate amounts of TNF-α, IL-6, IL-10 in the splenocyte supernatants, among high levels of TNF-α, IL-6 were considered as associated with survival in the challenged mice [29]. Notably, immunization with the A-G-eG_N_ and G-eG_N_ vaccine was capable of eliciting higher numbers of IFN-γ and IL-4 SFCs and level of cytokines compared with the numbers observed for A-G-eG_C_, G-eG_C_, A-G-NAb and G-NAb vaccinated mice (Figure 6A,B), which may be one reason why the mAbs of G_N_ has a better protective effect than those of G_C_ [5], which is worthy of further research. 

In the previous subunit vaccines [21], two-dose immunizations elicited high levels of neutralizing antibodies (1:5120), and delayed the death but could not produce enough protection to fight off viral infection in STAT129, the authors held that the high susceptibility of STAT129 mice to virus infection may result in under valuing the vaccine potential of experimental vaccines. However, except for mouse models, we also have different opinions, namely there were no relevant descriptions about a systemic immune response, namely specific antibody, a balanced Th1 and Th2 and T-cell immunity response in previous study that could be concerned with offering protection against lethal challenges [16,17,18,20,25,54]. In contrast to most other CCHFV vaccine studies, the study differed in that we performed IgG2a vs IgG1-specific ELISAs on samples collected 2 weeks after every immunization (Figure 4D–F). The data were shown that two vaccinations with 5 μg A-G-GP produced a dominant IgG1antibody response (Th2 immune response) and a delayed IgG2a antibody, suggesting that the Th1 immune response was really not as strong as Th2 immune response after two immunizations; a strong IgG2a antibody (Th1 response) was detected after three or four vaccinations in present study, which may be one reason why the previous protein subunit vaccines did not provide a protection. We also determined the ratio of IgG2a to IgG1 as an indirect method to evaluate induced Th1 and Th2-type biases, respectively, the result was shown that as the number of immunizations increased, all the ratios of IgG2a/IgG1 increased in 5 μg A-G-GP and 5 μg G-GP alone group, suggesting a Th1 immune response strengthened gradually until a balanced Th1and Th2 immune responses appeared, which might contribute protective immunity to CCHFV [20]. Therefore, we thought additional immunizations were needed to be added in the previous study. In addition, in the cell culture based and the mouse brain derived inactivated vaccines [16] and mRNA vaccine [24], authors held that all the immunocompetent mice may have a more balanced response than the type I interferon receptor knockout (IFNAR^−/−^) mice and antibody response in immunocompetent mice may be more reliable than the interferon knock-out mice. Therefore, it is reasonable that we verified the immunogenicity of G-GP in BALB/c mice and it is important to continue efforts to develop alternative CCHF animal models that can be used to evaluate the efficacy of glycoprotein-based subunit vaccines. In the future, we will verify the efficacy of the vaccine in suitable animal models, if the result show that the G-GP vaccines can achieve protection, additional experiments, such as passive antibody and T cell transfer assays are required to demonstrate which of the antibody responses and/or CD4+, CD8+ T cell after vaccination/infection are responsible for surviving from CCHFV infection.

In a word, we successfully developed three G-GP vaccine candidates displaying the GP subunits in this study. Our results clearly demonstrated that G-GP with ISA 201VG plus Poly (I:C) (A-G-GP) could elicit systemic immune responses in BALB/c mice, including specific humoral, T-cell and a balanced cell-mediated Th1 and humoral-mediated Th2 immune response. Therefore, the GEM-PA delivery system is promising for the development of subunit vaccines against CCHFV. In addition, it is the first report regarding a subunit vaccine expressing the conserved neutralizing antibody (NAb) region of CCHFV with good antigen-specific immunogenicity in BABL/c mice. The effectiveness of a G-GP vaccine against CCHFV has been severely hampered due to the lack of a suitable animal model and the requirement of a high-containment laboratory to handle the virus; we, along with others, recently developed small adult-animal models to test the protective effects and longevity, and even investigated whether the G-GP vaccine could offer cross-protection against multiple CCHFV strains in the future.

## 5. Conclusions

GEM-PA surface display system could provide an efficient and simple purification method for CCHFV subunit antigens and G-GP produced systematic humoral and cellular immune responses, which has the potential to be developed into a promising candidate vaccine against CCHFV infections.

## Figures and Tables

**Figure 1 viruses-14-01664-f001:**
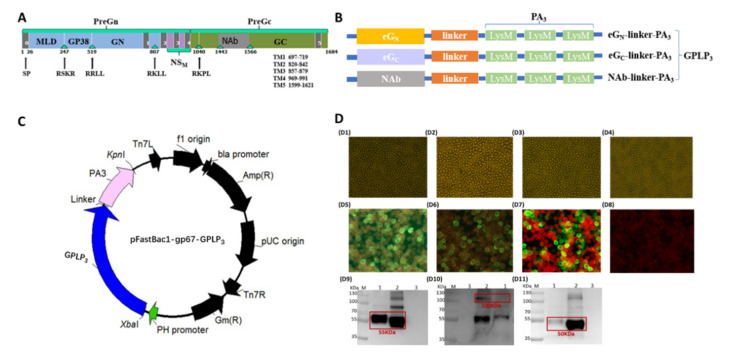
(**A**) Schematic representation of the glycoprotein ORF of CCHFV Ibar10200 strain and construction of the fusion proteins. The precise location of the signal peptide was predicted at aa 1 to 26 by the prediction server Signal P V1.1 (22). TM helix meaning transmembrane helix, and inside meaning the cytoplasmic side, the precise locations of the various regions of glycoprotein ORF predicted by the TMHMM server are as follows: aa 1 to 696 (outside), aa 697 to 719 (TM helix), aa 720 to 819 (inside), aa 820 to 842 (TM helix), aa 843 to 856 (outside), aa 857 to 879 (TM helix), aa 880 to 968 (inside), aa 969 to 991 (TM helix), aa 992 to 1598 (outside), aa 1699 to 1621 (TM helix), and aa 1622 to 1684 (inside). The tetrapeptide RRLL (aa 516 to 519) precedes the confirmed N terminus of G_N_, and RKPL (aa 1037 to 1040) precedes the confirmed N terminus of G_C_. The triangular symbols denote potential cleavage recognition sites: RSKR, aa 244 to 247 and RKLL, aa 804 to 807. The precise location of G_N_ ectodomain was predicted at aa 520 to 696, Gc ectodomain was predicted at aa 1041 to 1598, the truncated the protein by 19 carboxy-terminal amino acids, corresponding to the membrane-proximal stem region (FFYGLKNMLSGIFGNVFMG) was predicted at aa 1041 to 1579. NAb is a conserved neutralizing epitope of CCHFV glycoprotein was predicted at aa 1443 to 1566. (**B**) Schematic illustration of eG_N_-linker-PA_3_, eG_C_-linker-PA_3_, and NAb-linker-PA_3_ fusion proteins. (**C**) Schematic of the recombinant baculovirus expressing the CCHFV GPLP_3_ fusion protein. (**D**) CPE and detection of the expression of eG_N_-linker-PA_3_, eG_C_-linker-PA_3_, and NAb-PA_3_ fusion proteins. D1–D4: CPE of Sf9 cells were infected with recombinant baculoviruses rBV-eG_N_LP_3_, rBV-eG_C_LP_3_, rBV-NAbLP_3_, uninfected cells. D5–D8: Expression of eGPLP_3_ (Sf9 were infected with eG_N_LP_3_, eG_C_LP_3_, NAbLP_3_ or rBV) was evaluated by IFA using a mouse anti-CCHFV G_N_ polyclonal antibody or mouse anti-CCHFV G monoclonal antibody (magnification of microscopy images, 200×). D9–D11: Western blot analysis of the eG_N_LP_3_, eG_C_LP_3_, NAbLP_3_ or rBV protein expression in Sf9-infected cells. Expression was detected with a mouse anti-CCHFV G_N_ polyclonal antibody or mouse anti-CCHFV G_C_ monoclonal antibody. All target proteins are boxed out in red. (**A**) eG_N_LP_3_ was present as a 55 kDa protein.; (**B**) eG_C_LP_3_ was present as a 100 kDa protein; (**C**) NAbLP_3_ was present as a 50 kDa protein. M: molecular weight marker, 1: culture supernatant, 2: cell sedimentation, 3: rBV infected cells.

**Figure 2 viruses-14-01664-f002:**
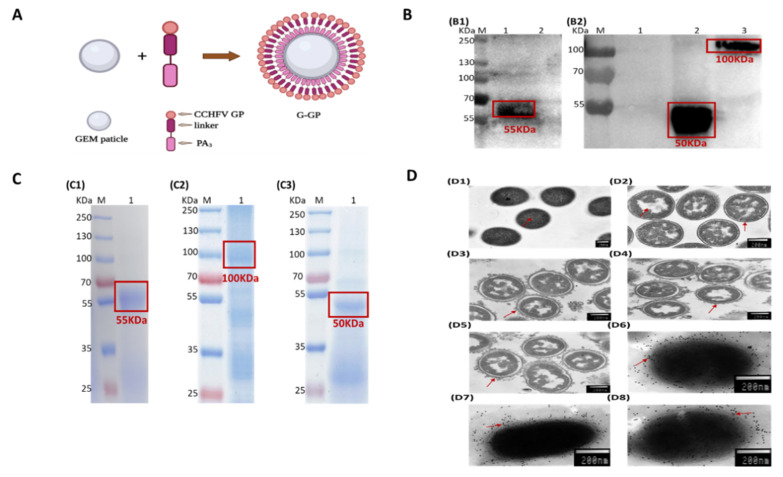
(**A**) Schematic representation of GP-PA binding with GEM particles. The CCHFV GP fragment of interest was fused at its C terminus with three LysM motifs (PA_3_). The PA bound noncovalently to the peptidoglycan to formulate G-GP antigen. (**B**) Western blot analysis of the proteins displayed on GEM particles. Lane B1–1, Lane B2–2, Lane B2–3: eG_N_LP_3_, NAbLP_3_, eG_C_LP_3_ proteins on GEM particles, proteins were detected with a mouse anti-CCHFV-G_N_ polyclonal antibody or a mouse anti-CCHFV-pre-Gc monoclonal antibody (11E7); Lane B1-2, Lane B2-1: the mock control. (**C**) Detection of the fusion proteins displaying the GEM particles. C1-C3: SDS-PAGE analysis of the displaying of eG_N_LP_3_, eG_C_LP_3_ and NAbLP_3_ proteins on GEM particles. Lane C-1, C-3: GEM particles displaying eG_N_LP_3_/ NAbLP_3_ from the culture supernatant of rBV-eG_N_LP_3_-infected Sf9 cells; Lane C-2: GEM particles displaying eG_C_LP_3_ from the supernatant after ultrasonic crushing of rBV-eG_C_LP_3_-infected Sf9 cells; (**D**) GEM particles identified by thin sectioning and immunoelectron microscopy of bacterium-like particles. D1: Untreated MG1363; D2: GEM particles; D3–D5: (G-eG_N_, G-eG_C_, G-NAb) particles; D6–D8: Immunoelectron microscopy of (G-eG_N_, G-eG_C_, G-NAb) particles.

**Figure 3 viruses-14-01664-f003:**
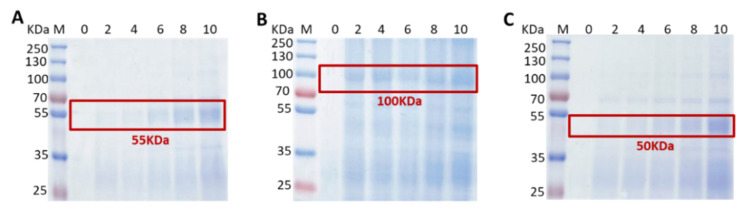
Detection of binding activity of the fusion proteins on GEM particles by SDS-PAGE. (**A**–**C**) The maximum binding capacity of each fusion protein binding to the GEM particles when 0.5 U GEM particles were combined with 0, 2, 4, 6, 8 and 10 mL of each recombinant baculovirus culture supernatant. All target proteins are boxed out in red. (**A**) G-eG_N_ was present as a 55 kDa protein; (**B**) G-eG_C_ was present as a 100 kDa protein; (**C**) G-NAb was present as a 50 kDa protein. M: molecular weight marker; 0: GEM particles.

**Figure 4 viruses-14-01664-f004:**
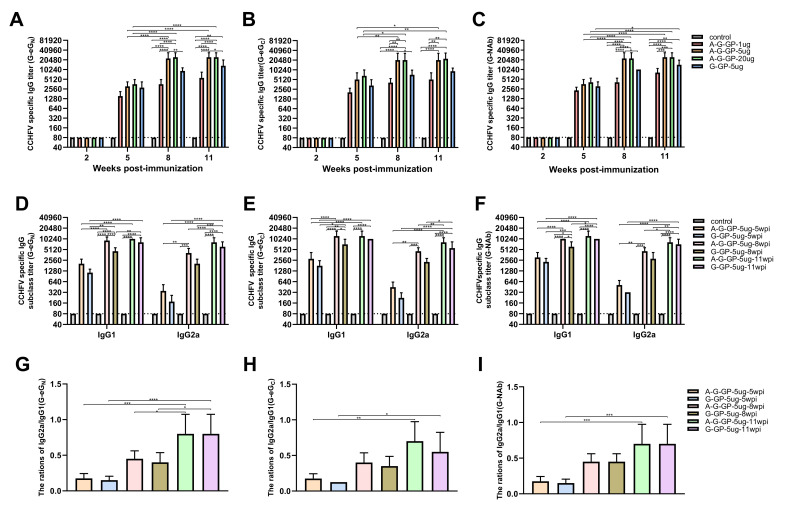
Serum antibody responses induced by CCHFV G-GP. Serum samples were collected by retro-orbital plexus puncture at weeks 2, 5, 8 and 11. CCHFV GP-specific total IgG (*n* = 5 mice/group/time point), IgG1and IgG2a (*n* = 5 mice/group/time point) antibody responses were measured by indirect ELISA with the purified eG_N_ protein and eG_C_ and are displayed as the end-point dilution titers. The horizontal dotted line in the figure indicates the limit of determination (LOD). (**A**–**C**) Analysis of serum antibody titers induced by different doses and time points on the immune system in mice by ELISA, data are shown as the mean ± SD and were analyzed by two-way ANOVA; (**D**–**F**) Analysis of serum antibody subclass titers induced by time points on the immune dose (5 μg G-GP) by ELISA, data are shown as the mean ± SD and were analyzed by one-way ANOVA; (**G**–**I**) Analysis of serum antibody ratios of IgG2a/IgG1 induced by time points on the immune dose (5 μg G-GP) by ELISA, data were shown as the mean ± SD and were analyzed by one-way ANOVA, * *p* < 0.05, ** *p* < 0.01, *** *p* < 0.001, **** *p* < 0.0001.

**Figure 5 viruses-14-01664-f005:**
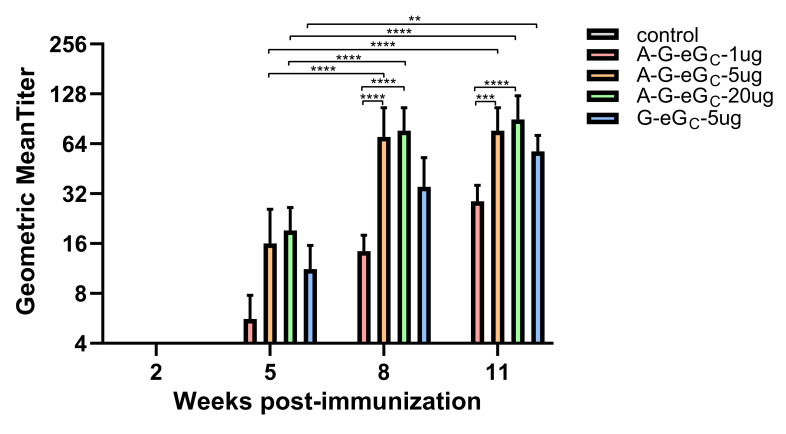
Analysis of serum antibody titers induced by different immunizing doses by neutralization of CCHFV GP-pseudo-typed virus. Neutralizing antibody titers were measured with Vero E6 cells and 200TCID_50_ of pseudo-typed virus. Serum samples were collected 2 weeks after every immunization. Data are shown as the mean ± SD and were analyzed using one-way ANOVA (** *p* < 0.01, *** *p* < 0.001, **** *p* < 0.0001).

**Figure 6 viruses-14-01664-f006:**
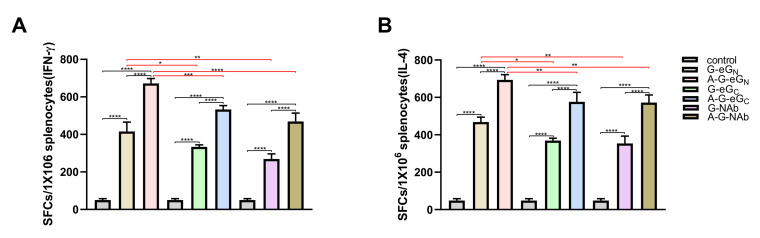
ELISpot analysis of IFN-γ, IL-4 secretion by mouse splenocytes (**A**,**B**). The splenocytes were collected from each group 14 days after the third immunization treated and restimulated with eG_N_ or NAb (10 μg/mL). Data are shown as the mean ± SD and were analyzed using one-way ANOVA (* *p* < 0.05, ** *p* < 0.01, *** *p* < 0.001, **** *p* < 0.0001).

**Figure 7 viruses-14-01664-f007:**
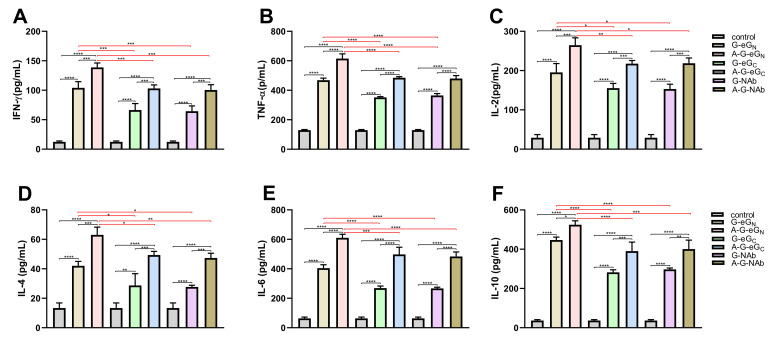
Detection of cytokine secretion levels in splenocytes that were restimulated with eGN or NAb (10 μg/mL and cultured in vitro. The secretion of (**A**) IFN-γ, (**B**) TN-α, (**C**) IL-2, (**D**) IL-4, (**E**) IL-6, (**F**) IL-10 was measured by using ELISA kit. Data are shown as the mean ± SD and were analyzed using one-way ANOVA (* *p* < 0.05, ** *p* < 0.01, *** *p* < 0.001, **** *p* < 0.0001).

**Table 1 viruses-14-01664-t001:** Oligonucleotide primers used in this study.

Oligonucleotide Primers Sequence (5’–3’) Enzyme Site		
eG_N_-F ^1,3^	TGCTCTAGA**CATCACCATCACCATCAC**TCTGAGGAACCTAGCGACG	XBaI
eG_N_-R	ACCAGAACCACCACCAGAACCACCCAGGAAGGCCATGGTGGTCT	
eG_C_-F ^1,3^	TGCTCTAGA**CATCACCATCACCATCAC**TTCCTGGACTCCACCGCC	XBaI
eG_C_-R	ACCAGAACCACCACCAGAACCACCGCTCTTCACGGATTCCAGCCAAC	
NAb-F ^1,3^	TGCTCTAGA**CATCACCATCACCATCAC**TCCGGCCTGAAGTTCGC	XBaI
NAb-R	ACCAGAACCACCACCAGAACCACCACAGGTGGAGTTCTGCTTGCC	
PA_3_-F ^2^	GGTGGTTCTGGTGGTGGTTCTGGTGATGGTGCTTCTTCAGCTGGT	
PA_3_-R ^1^	CGGGGTACCTTACTTGATACGCAGGTATTGACC	KpnI

^1^ Restriction enzyme sites are underlined and italicized. ^2^ The middle linker (Gly-Gly-Ser-Gly) × 2 base sequences are underlined. ^3^ His-tag base sequences are in bold.

**Table 2 viruses-14-01664-t002:** The mouse vaccination protocols.

Group	*n*	Immunization Route	Antigen	Adjuvant
		A-G-eG_N_	15	subcutaneous	1/5/20 μg A-G-eG_N_	201VG + Poly(I/C)
		A-G-eG_C_	15	subcutaneous	1/5/20 μg A-G-eG_C_	201VG + Poly(I/C)
		A-G-NAb	15	subcutaneous	1/5/20 μgA-G-NAb	201VG + Poly(I/C)
		G-eG_N_	5	subcutaneous	5 μg G-eG_N_	-
		G-eG_C_	5	subcutaneous	5 μg G-eG_C_	-
		G-NAb	5	subcutaneous	5 μg G-NAb	-
	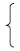	GEM + 201 + polyI:C	5	subcutaneous	GEM + 201 + polyI:C	-
Control	201 + polyI:C	5	subcutaneous	201 + polyI:C	-
	PBS	5	subcutaneous	PBS	-

**Table 3 viruses-14-01664-t003:** Comparison of the serological responses in BALB/c mice subcutaneously immunized with G-GP vaccines against CCHFV. Booster doses were given at 3, 6 and 9 weeks after the first immunization, and virus-specific antibodies in mice were assessed at 2, 5, 8, 11 weeks after the first immunization. The results were recorded as the geometric mean titer (GMT) ± the standard error (S.D.).

Vaccine	Dose (μg)	Antibody Titer (GMT)							
		First Immunization		Second Immunization		Third Immunization		Fourth Immunization	
		ELISA	N ab	ELISA	N ab	ELISA	N ab	ELISA	N ab
G-eG_N_	20	ND	ND	3584 ± S.D	ND	24576 ± S.D	ND	24576 ± S.D	ND
	5	ND	ND	3072 ± S.D	ND	22528 ± S.D	ND	24576 ± S.D	ND
	1	ND	ND	1536 ± S.D	ND	3584 ± S.D	ND	5632 ± S.D	ND
G-eG_C_	20	ND	ND	1126.4 ± S.D	19.2 ± S.D	20480 ± S.D	76.8 ± S.D	22528 ± S.D	89.6 ± S.D
	5	ND	ND	5120 ± S.D	16 ± S.D	20480 ± S.D	70.4 ± S.D	20480 ± S.D	76.8 ± S.D
	1	ND	ND	2048 ± S.D	5.6 ± S.D	4096 ± S.D	14.4 ± S.D	5120 ± S.D	28.8 ± S.D
G-NAb	20	ND	ND	4096 ± S.D	ND	22528 ± S.D	ND	24576 ± S.D	ND
	5	ND	ND	3584 ± S.D	ND	22528 ± S.D	ND	24576 ± S.D	ND
	1	ND	ND	2304 ± S.D	ND	4096 ± S.D	ND	8192 ± S.D	ND

GMT: The end point antibody titer represented as the mean ± S.D. of five animals. N ab; Neutralization antibody titer. ND: Not determined.

## Data Availability

Not applicable.

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
