# Peer review of "GEM-PA-Based Subunit Vaccines of Crimean Congo Hemorrhagic Fever Induces Systemic Immune Responses in Mice"

_viruses, 2022, doi:10.3390/v14081664_

Round 1

Reviewer 1 Report

In this manuscript, Wang et al., developed subunit vaccine against CCHFV using bacterial like particles heterologous protein expression system. Overall, the study is well designed, and the appropriate experiments are done to support the study. Although, data is convincing but there are points to be considered by the authors.

1.       Manuscript is hard to follow so thorough rephrasing of the manuscript text is highly recommended.

2.       Authors are advised to shorten the abbreviations (GPBLP3, eGNBLP3 …etc.) used throughout the manuscript.

3.       Abstract:

Line-23: Include a statement about the significance of GEM-PA surface display system over other vaccine development system.

Line-24: ‘three subunit vaccines….were developed’ In this study, authors have not developed three different vaccines; instead a GEM-PA based vaccine expressing three subunit vaccine candidates of CCHFV was developed. Authors are requested to correct this and if this is not the case please specify properly.

4.       Introduction:

Line 61: Change CCHFV behaves a wide range of hosts to CCHFV infects a wide range of hosts

Line 63: It should be asymptomatic rather than a symptomatic.

Line 76: Replace ‘people’ to ‘humans’

Line 85: Replace ‘purity product’ to ‘product purity’

Line-86: Authors are suggested to include the reason of opting GEM-PA surface display system over other vaccine development system.

Line 90: Rephrase

Line-103: Simplify the difference between GPLPA3 complexes and GPBLP3

Line 104-110: Rephrase

5. Materials and Methods:

2.1: This section is hard to follow. Authors are advised to include a schematic for better understanding of the readers.

2.5: Line 221: Simplify A-GPBLP3

6. Results:

Line-344: Asterisks are missing.

Fig. 1C:  The schematic should be the part of Fig. 2 and authors are suggested to use three different colors representing three different antigens.

3.3, Line 398, Fig. 3B: For Gc , obvious increase in the binding with increase in supernatant volume is not visible. Please comment.

Fig. 3 legend, Line 406, Please indicate the protein specific gel earlier. 

Reviewer 2 Report

Thank you for the opportunity to review the submitted manuscript entitled “Bacterium-Like Particle-Based Subunit vaccines of Crimean Congo Hemorrhagic Fever Induces Systemic Immune Responses in Mice” by Wang et al.

The manuscript describes the production of baculovirus expressed protein subunit vaccines against CCHF and immunogenicity in BALB/c mice. I suggest the manuscript is revised for English language. I have put some examples below.

Secondly the title ‘Bacterium-Like Particle-Based’ could be confusing as the material is produced in baculovirus system and then assembled on a gram-positive enhancer matrix (GEM) scaffold, perhaps GEM nanoparticles would be more appropriate.

The manuscript describes the immune response in Balb/C mice to the particles but does not go as far as challenging immunised mice, this would require high containment facilities and understand that this is not within the scope of the manuscript but would be a great next step for this work, it would therefore also be interesting to see the immunogenicity in a CCHFV susceptible mouse strain such as interferon deficient A129 model.

The discussion highlights the discordance between CCHFV neutralisation and ELISA assays which could be expanded in respect to issues around vaccine development for CCHFV.

The manuscript does describe a technology platform applied to CCHFV vaccinology and is an interesting area that deserves highlighting to your readership and I therefore recommend publication with modifications.

Revision of English examples:

Line 57: please revise, ‘transmit to humans’

Line 63: change a symptomatic to asymptomatic, the end of the sentence also requires clarification e.g. ‘even with high viremia’.

Line 64: please revise sentence end ‘tested in human aged over 16 years of since 1974 in Bulgaria’ for clarity.

Line 69: delete word has

Line 73: please revise sentence.

Line 90: change leaved to remained

Line 98: Please revise for English to ‘parasitic, viral or bacterial antigens’

Lines 122, 128, 139, 141, 143: the word respectively is used multiple times and in most cases not required for understanding.

Line 126: change sites to site

Line 137: Change 4day to 4 days

Line 566: remove ‘were’

Other comments:

Line 414: Confirm dosing, other text refers to 1,5,20 not 5,10,20.

Figure 4 y-axis G/H/I: change Rations to Ratios
